# Predictors of Entrepreneurial Intention among High School Students in South Korea

Min-Sun Kim [1], Andrian Dolfriandra Huruta [2,*] and Cheng-Wen Lee [3]

[1] International Undergraduate Program in Business and Management, Chung Yuan Christian University, 200 Zhong Bei Rd, Taoyuan City 320314, Taiwan
[2] Department of Economics, Satya Wacana Christian University, 52-60 Diponegoro Rd, Salatiga 50711, Indonesia
[3] Department of International Business, College of Business, Chung Yuan Christian University, 200 Zhong Bei Rd, Taoyuan City 320314, Taiwan
* Correspondence: andrian.huruta@uksw.edu

**Abstract:** According to the theory of planned behavior, the goal of this research is to evaluate the connections between perceived entrepreneurial capacity, perceived social norm, attitude toward entrepreneurship, and entrepreneurial intention. It also examines the mediating effect of the educational environment on these connections. Based on the conceptual structure of the theory, this research study was organized to explore how the conception of the theory works in the case of Korean students and to further assess the role of the educational environment in the course of its operation. At Mirae High School of Science and Technology, eighty-nine respondents were used for the statistical analysis. One model was chosen to explore this research concept, namely, variance-based structural equation modeling (VB-SEM). All of the direct hypotheses are supported except the indirect hypothesis. The perceived entrepreneurial capacity positively influences entrepreneurial intention. The perceived social norm positively influences entrepreneurial intention. Additionally, a positive attitude toward entrepreneurship influences the intention to engage in it. The indirect effect is not in line with the expected hypothesis. The research's findings contribute to the literature review by adding another empirical confirmation (educational environment's role) from South Korean students' viewpoint. It offers useful information and provides students with entrepreneurial skills. This research also contributes to the advancement of knowledge in relation to the predictors of entrepreneurship in students, and it could help governments to make decisions on entrepreneurship. Moreover, it highlights the direction in which a government or policymaker can take to pursue entrepreneurship and its education.

**Keywords:** entrepreneurial intention; theory of planned behavior; educational environment; South Korea; VB-SEM

## 1. Introduction

There is currently much discussion about entrepreneurship. The question of how to develop entrepreneurship has engaged current scholarship across various fields. The definition of entrepreneurship is starting a new business with innovative ideas [1,2]. Entrepreneurship has some economic virtues: innovation, economic growth, and job creation [1,3]. It is worth observing what factors elicit entrepreneurship. The start of an entrepreneurial venture is firmly wedded to exploring opportunities in an entrepreneurial context [1,4]. Identifying opportunities does not occur by itself, and this ability can be fostered by particular conditions [1–3]. However, entrepreneurship generally has its own risk in its performance [5]. Thus, with the ability to identify opportunities, the entrepreneurial intention of realizing the entrepreneurial opportunities is needed for entrepreneurship [1,6–11]. Intention allows certain behavior, and several factors constitute the intention [12,13]. The mental desire to try is as much a powerful motive as the ability to perform things [12].

The theory of planned behavior states that perceptions of behavioral control, subjective norms, and attitude toward the behavior all affect behavioral intention [6–12,14]. Based on the conceptual structure of the theory, this research is organized to explore how the conception of the theory works in the event of Korean students. The Korean government has recognized the economic potential of entrepreneurship and has set various policies to boost entrepreneurship [15,16]. Among different economic policies, start-up promotion has been considered as one solution capable of breaking through economic difficulties such as youth unemployment and unbalanced growth [15]. The Korean government has promoted startups with the Ministry of SMEs and Startups (MSS) as its center. Its implementation incorporates efforts to nurture future entrepreneurs [15]. This research was carried out at Mirae High School of Science and Technology in South Korea. The artificial intelligence program at Mirae High School of Science and Technology is driving the fourth industrial revolution. At the school, students are instructed in the study of invention and artificial intelligence. This school has received support from the Korean Intellectual Property Office and the Seoul Metropolitan Office of Education. Since 2010, the school has placed a strong emphasis on innovative education, using original educational invention materials such as research science for a given product. As entrepreneurship takes on economic importance, the education specialized for nurturing entrepreneurship has been focused. Recent years have seen an increased discussion on entrepreneurship education from various aspects. Many related research papers have been dominated by a focus on undergraduate or graduate business education [17–19], and this paper considers the educational environment as a mediator in the existing structure of the theory of planned behavior. Given these considerations, this research is certainly welcome.

The goal of this study is to assess the connections between perceived entrepreneurial capacity, perceived social norm, attitude toward entrepreneurship, and entrepreneurial intention using the theory of planned behavior. It also examines the mediating effect of the educational environment on these relationships. This research will provide a useful compendium on the ways in which each factor affects entrepreneurial intention and the role of the educational environment in a Korean context. This research highlights how government policymakers should approach entrepreneurship and its education.

This research is organized into several parts. An overview of the importance of entrepreneurship is given at the beginning of the research. A literature review and the development of a hypothesis are then presented in Section 2. The research methodology is presented in Section 3. Section 4 of the research presents empirical findings. Section 5 of the research discusses the findings. The conclusion (which includes limitations and offers recommendations) is also provided to support the entrepreneurial intentions in Section 6.

## 2. Literature Review

### 2.1. Theory of Planned Behavior

Intention is a cardinal factor in the theory of planned behavior. An individual's intention is a driver of behavior while being an indicator of other motivational factors. In general, the willingness to perform a given behavior and its planning to one's best efforts constitutes behavioral intention [7]. However, volitional control over given opportunities and resources serves as an indispensable factor for the behavioral intention [7,9]. The ability to control those environmental factors elevates the probability of behavioral achievement. Not only does actual control increase the likelihood, the perception of one's ability to control environmental factors also offers a positive influence on intention and behavior [7,9]. A person who believes that the one has neither resources nor opportunity is unlikely to have a behavioral intention [9]. Subjective norms are a person's perception of what those close to them think about that person's behavior [7,9]. It is a normative belief of what an individual is allowed to do in one's society [7,9,20]. The perception of how a person matches their social standard affects their intention [7,9,20]. An approval of important referent people acts as a significant influencer of intention [7]. A positive assessment of a particular behavior results in a favorable attitude toward the behavior [9,21], and favorability constitutes a

motivational factor of trying [8,20]. According to the theory of planned behavior, attitudes toward the behavior, subjective norms, and perceived behavioral control all play a role in intention [2,6–9,14,22], and each factor influences intention independent of the other two factors [9].

## 2.2. Entrepreneurial Intention

Intention is one's mental state as it relates to engaging oneself into performing a certain behavior [20]. A strong intention equips person for behavior, and the intention is formed by diverse factors [8]. Entrepreneurship does not take place accidentally [23]. It is the activity arisen from one's choice [23]. An attitude known as entrepreneurial intention encourages entrepreneurial behavior [2,3,22–24]. Enough has been said to demonstrate that entrepreneurial intention serves as a direct and powerful driver of entrepreneurship [23]. Obschonka et al. noted that entrepreneurial competence nurtured in one's childhood and adolescence leads to the development of an entrepreneurial mind in one's adulthood [23]. Entrepreneurial intention is explained by both personal and social factors [2,6,7,10,11,14,25,26]. Certain personality traits or demographic characteristics are significantly associated with entrepreneurial intention [26]. Entrepreneurship is not an activity which instantaneously takes place when someone has decided to begin their own startup but rather is an activity which takes time to develop [27]. It needs certain personal characteristics to successfully exploit an entrepreneurial chance when it comes. These characteristics cannot form in a short time [27]. Gartner suggested that environmental factors as well as a certain personality enable entrepreneurship [27]. Regardless of whether it is a material or mental factor, a certain environmental factor which is supportive of entrepreneurship takes on the role of a stimulus in entrepreneurship [25,27,28].

## 2.3. Perceived Entrepreneurial Capacity

Identifying entrepreneurial opportunity involves a perceived level of uncertainty in the environment. The perceived ability of controlling opportunities and resources affects entrepreneurial intention [28]. A high level of perception of competence in achieving a goal comes from a positive judgement about the situation which a person faces. If the judgment about one's entrepreneurial competence is negative, the individual could have less entrepreneurial intention [29]. Tsai et al. noted that a lower entrepreneurial intention is manifested by fear of failure [29]. It acts as an obstacle to realizing entrepreneurship [10,11,29]. Overcoming the fear of failure in one's entrepreneurship is instrumental to building confidence in entrepreneurship. The self-efficacy in entrepreneurship is in line with the confidence. Another way to express highly perceived entrepreneurial capacity is through the concept of self-efficacy in business [29]. The cognition of self-efficacy is positively correlated with entrepreneurial intention [22,30]. Perceived entrepreneurial capacity is attributed not only to human capital factors but also to acquired resources from one's given situation. Skill, knowledge, and experience from one's career and entrepreneurship-related learning allow for the development of entrepreneurial competence; moreover, the consequential increase in a sense of self-efficacy propels entrepreneurial intention [31]. Self-efficacy is a crucial characteristic in exercising entrepreneurial tasks, and entrepreneurial intention is linked to one's ability [32]. At this point, this research can develop a hypothesis as follows:

**Hypothesis 1 (H1).** *Entrepreneurial intention is positively impacted by perceived entrepreneurial capacity.*

## 2.4. Perceived Social Norm

The idea developed from social relationship affects entrepreneurial intention [10,11,24]. The point of view of family or friends serves as a benchmark in selecting career. Dragin et al. found that family characteristics and tradition greatly affect one's entrepreneurial intention [33]. If financial support as well as sentimental support is expected from family

or friends when someone starts a startup, the impact in selecting an entrepreneurial career could be much stronger than when there is no financial support from them [24]. In that sense, it is highly probable that one's perceived social norm affects their entrepreneurial intention. Social norm is an individual's perception of other's opinions about a particular behavior. Entrepreneurial value arises from an exposure to the supportive surroundings of entrepreneurship and an interaction with people who have entrepreneurial spirit. A favorable social support strengthens one's willingness to become an entrepreneur [34]. Social interaction with people who are in favor of entrepreneurship and an accessibility to resources are two supportive factors which facilitate entrepreneurial intention [25]. Educational settings for improving entrepreneurial competence and its educators also serve to develop a favorable social norm to entrepreneurship [2,22,25]. One's entrepreneurial intention is dictated by how a person perceives their surroundings and opinions in terms of entrepreneurship [25]. Sobaih and Elshaer found that gender roles established from social custom also determines the forming of favorable characteristics that are conducive to entrepreneurial intention [35]. At this point, this research can develop a hypothesis as follows:

**Hypothesis 2 (H2).** *Entrepreneurial intention is positively impacted by perceived social norm.*

### 2.5. Attitude toward Entrepreneurship

A certain attitude allows for an understanding of the intention toward a given behavior [8,12]. Some personality traits (extraversion, conscientiousness, agreeableness, openness to experience, and so on) are linked to entrepreneurship, considering that career is a vehicle of self-realization [23]. An individual's attitude toward entrepreneurship shows the level of desire to become an entrepreneur and the confidence that they will succeed [10,11,25]. Boubker et al. found that one's attitude toward entrepreneurship has a positive and significant effect on entrepreneurial intention [2,22]. Moreover, many research papers have suggested that attitude toward entrepreneurship serves as a key determinant of entrepreneurial intention [2,22,36,37]. At this point, this research can develop a hypothesis as follows:

**Hypothesis 3 (H3).** *Entrepreneurial intention is positively impacted by one's attitude toward entrepreneurship.*

### 2.6. Educational Environment

Entrepreneurial competence in one's childhood and adolescence are positively correlated with engagement in entrepreneurship. The manifestation of entrepreneurship is retroactive to early entrepreneurial competences. Entrepreneurship is expected of the person who experienced early commercial activities, leadership, and early inventions in one's childhood and adolescence [23]. Boubker et al. found that entrepreneurial intention among students is positively and significantly impacted by entrepreneurship education. Entrepreneurship education serves as a catalyst for the development of students' entrepreneurial intentions. Students' attitudes toward building a startup contribute to their schools' intention of nurturing entrepreneurial intention. There are enough research papers to prove that entrepreneurial education facilitates students' intentions to begin a startup [2,22]. Moreover, entrepreneurial training involves developing entrepreneurial skills and entrepreneurial intention [2,10,11,22,23]. Some entrepreneurial personal traits (proactiveness, innovativeness, and risk-taking) can be gained from entrepreneurial training [2,22]. Accordingly, an entrepreneurial education environment is supportive of understanding students' entrepreneurial intention [2,22,38]. Aliedan et al. found that undergraduate education supports building entrepreneurship. They noted that entrepreneurial education is vital to the development of entrepreneurial intention in both a direct way and an indirect way. This shows that entrepreneurial intention hinges on the level of education and its curriculum [17]. Chang et al. found that entrepreneurial behavior originated from

intention, mindset, and competency, and that these three are developed in an educational setting [39]. Many recent research papers prove educational role in forming entrepreneurship [40]. Recent scholarly effort has shown that entrepreneurship cannot be realized without an educational setting [41]. The attitude toward the behavior, subjective norms, and perceived behavioral control, according to the theory of planned behavior, have an impact on behavioral intention [2,6–9,14,22]. This research identified the educational environment as a mediating factor within the theory of planned behavior to explore the role of education in the relationship between entrepreneurial intention and three factors: perceived entrepreneurial capacity, perceived social norm, and attitude toward entrepreneurship. At this point, this research can develop a hypothesis as follows:

**Hypothesis 4 (H4).** *The relationship between perceived social norms, entrepreneurial attitudes, and entrepreneurial intention is indirectly influenced by the educational environment.*

## 3. Research Methodology

### 3.1. Research Design and Sample

This cross-sectional study was carried out in South Korea. Students at the Mirae High School of Science and Technology participated in a survey. The survey period was conducted from August to September 2021. All participants were students aged between 15 and 19 years old. The research population involved 100 students. Purposive sampling was used in this study to gather the sample. The number of pointing arrows and the statistical power rule were used in this study to determine the minimum sample size [42]. The minimum sample size is 69 to achieve statistical power of 80% ($R^2$ value at least 0.5 and 1% probability of error).

The sample size was chosen in accordance with the Variance Based-SEM (VB-SEM) research's minimal sample size requirement [43]. A total of 89 samples were obtained because more than 11% (11 questionnaires) of the answers were missing or deemed invalid. Eighty-nine samples in total were used to analyze the data. In order to validate the research model and hypotheses, we used the SmartPLS 3 program.

### 3.2. Assessing Reflective Measurement Models

This research employs the reflective model. In VB-SEM or PLS-SEM terminology, the reflective model reveals that a change in the indicator is driven by changes in the latent variable. Reflective measurement models are the most widely used measurement models. All three are assessed: internal consistency reliability, convergent validity, and discriminant validity. Table 1 summarizes the pertinent assessment criteria and their desired values.

**Table 1.** Models for reflective measurement and evaluation.

| Criteria | Description | Test | Desirable Values | Reference |
|---|---|---|---|---|
| Internal consistency | The type of reliability used to evaluate a result's consistency between different items. The objective is to determine whether there is sufficient correlation among items to demonstrate that the items of the same latent variable are comparable. | Cronbach's alpha (CA) | 0.70 (satisfactory) to 0.90 (good); 0.60–0.70 acceptable for exploratory research | [44] |
| | | rho $_A$ | 0.70 (satisfactory) to 0.90 (good); 0.60–0.70 acceptable for exploratory research | [45] |
| | | Composite reliability (CR) | 0.70 (satisfactory) to 0.90 (good); 0.60–0.70 acceptable for exploratory research | [46] |
| | | Confidence intervals | Minimum threshold above lower bound of the 95% bootstrap confidence interval | [47] |

**Table 1.** *Cont.*

| Criteria | Description | Test | Desirable Values | Reference |
|---|---|---|---|---|
| Convergent validity | Determines which items in the same latent variable have a positive correlation with other items. | Indicator Reliability (Outer Loadings) | >0.60 | [43] |
| | | Average Variance Extracted (AVE) | >0.50 | [48] |
| Discriminant validity | Determines whether a latent variable is measuring a distinct construct and how well the items reflect the target construct. | Bootstrapping | Bootstrapped confidence interval of HTMT should not contain 1.0 | [49] |
| | | Heterotrait–Monotrait (HTMT) ratio of correlations | <0.90 if constructs are conceptually similar, otherwise <0.85 | [50] |

### 3.3. Latent Variables and Indicators

This research has five main latent variables, namely, perceived entrepreneurial capacity (PEC), perceived social norm (PSN), attitude toward entrepreneurship (AE), educational environment (EE), and entrepreneurial intention (EI) (Table 2).

**Table 2.** Latent variables and indicators' formulation.

| Code | Latent Variables | Indicators/Items |
|---|---|---|
| PEC | Perceived Entrepreneurial Capacity | Entrepreneurial self-efficacy. |
| | | Internal locus of control for entrepreneurship. |
| | | Perception of competence for running a startup. |
| | | Perception of confidence in running a startup. |
| PSN | Perceived Social Norm | Family's approval for starting a startup. |
| | | Friends' approval for starting a startup. |
| | | Teachers' approval for starting a startup. |
| AE | Attitude toward Entrepreneurship | Perception of advantage of being an entrepreneur. |
| | | Perception of entrepreneur as a career. |
| | | Adventurous spirit for starting a startup. |
| | | Perception of satisfaction of being an entrepreneur. |
| EE | Educational Environment | Curriculum nurturing a creative idea. |
| | | Interactive learning environment. |
| | | Skills-based learning. |
| | | Inspiring creativity in class activities. |
| EI | Entrepreneurial Intention | Readiness to become an entrepreneur. |
| | | Willingness to do one's best to build a startup. |
| | | Intention of starting a startup in the future. |
| | | Confidence for starting a startup. |
| | | Goal setting for starting a startup. |
| | | Seriousness in building a startup. |

Source: Modified from Boubker et al. [2].

We provide one path diagram in Figure 1 below to illustrate the relationships between latent variables.

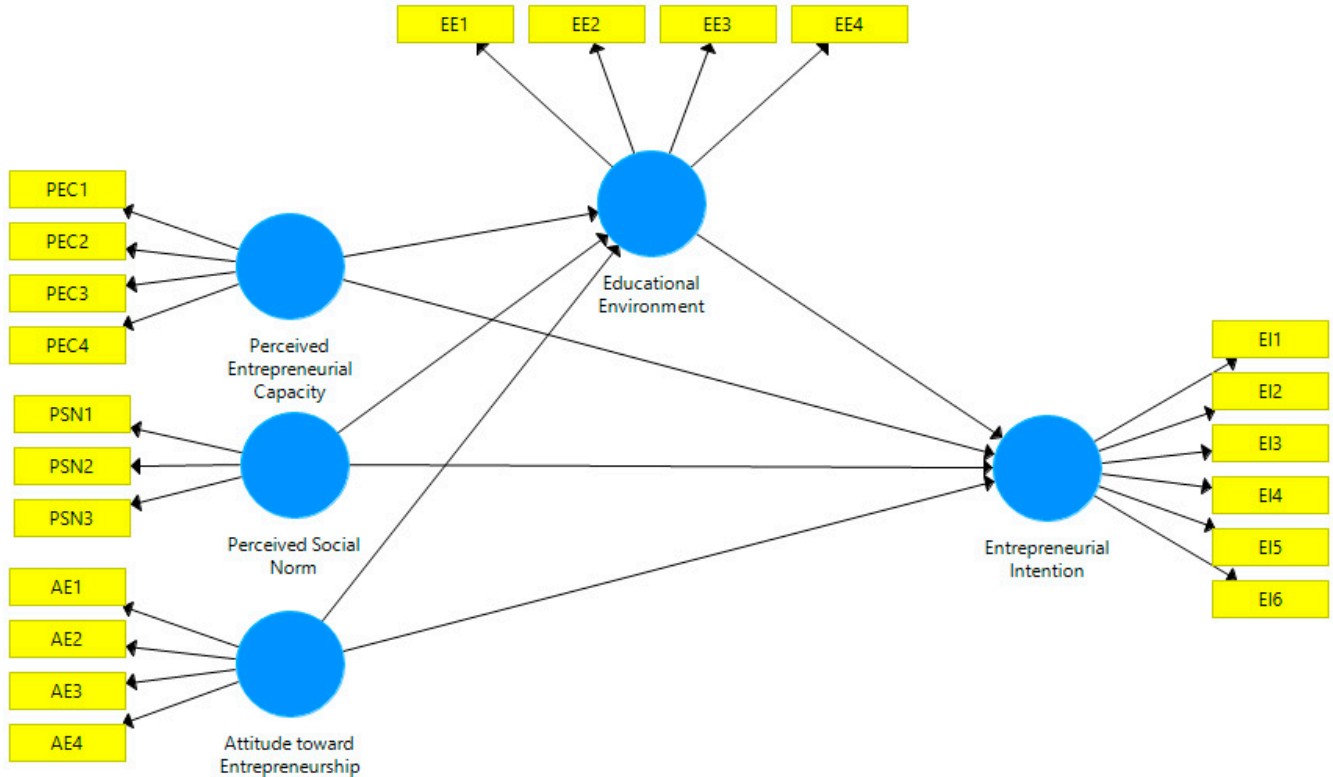

**Figure 1.** Research framework.

For the purpose of examining the predictors of students' entrepreneurial intention, the theory of the planned behavior model may be helpful. Figure 1 shows that perceived entrepreneurial capacity has four indicators (PEC1, PEC2, PEC3, and PEC4), perceived social norm has three indicators (PSN1, PSN2, and PSN3), attitude towards entrepreneurship has four indicators (AE1, AE2, AE3, and AE4), educational environment has four indicators (EE1, EE2, EE3, and EE4), and entrepreneurial intention has six indicators (EI1, EI2, EI3, EI4, EI5, and EI6).

## 4. Empirical Results

There were 60 male and 29 female respondents from four departments at Mirae High School of Science and Technology (Table 3).

**Table 3.** Respondents' characteristics.

| Gender | Frequency | Percent |
|---|---|---|
| Female | 29 | 32.6 |
| Male | 60 | 67.4 |
| Total | 89 | 100.0 |
| Age | Frequency | Percent |
| 15 | 14 | 15.7 |
| 16 | 37 | 41.6 |
| 17 | 19 | 21.3 |
| 18 | 10 | 11.2 |
| 19 | 9 | 10.1 |
| Total | 89 | 100.0 |

**Table 3.** *Cont.*

| Department | Frequency | Percent |
|---|---|---|
| Department of AI contents | 29 | 32.6 |
| Department of Visual Design | 14 | 15.7 |
| Department of Maker and Creation | 27 | 30.3 |
| Department of Computer and Patent | 19 | 21.3 |
| Total | 89 | 100.0 |
| Parent's Occupation | Frequency | Percent |
| Human Services | 18 | 20.2 |
| Engineering, Manufacturing, and Technology | 18 | 20.2 |
| Business, Management, and Administration | 48 | 53.9 |
| Health Science Technology | 3 | 3.4 |
| Agriculture, Food, and Natural Resources | 2 | 2.2 |
| Total | 89 | 100.0 |
| Family background in entrepreneurship | Frequency | Percent |
| Yes | 12 | 13.5 |
| No | 77 | 86.5 |
| Total | 89 | 100.0 |

Source: own research.

Most of our respondents were 16 years old, and their family occupation occurred mostly in the field of business, management, and administration. The absence of a family background in entrepreneurship became a fit target for the survey. Moreover, Table 4 shows the response category of indicators.

**Table 4.** Response category of indicators.

| Indicators | Mean | Median | Min. | Max. | $\sigma$ |
|---|---|---|---|---|---|
| PEC1 | 3.438 | 3 | 1 | 5 | 1.049 |
| PEC2 | 3.573 | 4 | 1 | 5 | 1.026 |
| PEC3 | 2.494 | 2 | 1 | 5 | 1.093 |
| PEC4 | 3.360 | 3 | 1 | 5 | 1.124 |
| Average | 3.216 | | | | |
| Category | Medium | | | | |
| PSN1 | 3.764 | 4 | 1 | 5 | 1.081 |
| PSN2 | 3.831 | 4 | 1 | 5 | 0.902 |
| PSN3 | 3.831 | 4 | 1 | 5 | 0.951 |
| Average | 3.809 | | | | |
| Category | High | | | | |
| EE1 | 3.719 | 4 | 1 | 5 | 1.005 |
| EE2 | 3.809 | 4 | 1 | 5 | 0.993 |
| EE3 | 3.854 | 4 | 1 | 5 | 0.978 |
| EE4 | 3.831 | 4 | 1 | 5 | 0.986 |
| Average | 3.803 | | | | |
| Category | High | | | | |

**Table 4.** *Cont.*

| Indicators | Mean | Median | Min. | Max. | σ |
|---|---|---|---|---|---|
| AE1 | 3.899 | 4 | 1 | 5 | 0.937 |
| AE2 | 4.18 | 4 | 1 | 5 | 0.906 |
| AE3 | 4.056 | 4 | 1 | 5 | 1.042 |
| AE4 | 3.978 | 4 | 1 | 5 | 1.038 |
| Average | 4.028 | | | | |
| Category | High | | | | |
| EI1 | 2.854 | 3 | 1 | 5 | 1.204 |
| EI2 | 3.865 | 4 | 1 | 5 | 1.073 |
| EI3 | 3.360 | 3 | 1 | 5 | 1.084 |
| EI4 | 2.753 | 3 | 1 | 5 | 1.202 |
| EI5 | 2.775 | 3 | 1 | 5 | 1.159 |
| EI6 | 2.798 | 3 | 1 | 5 | 1.317 |
| Average | 3.068 | | | | |
| Category | Medium | | | | |

Note: σ indicates the standard deviation. The response category is cited from Hair et al. [51], Atahau et al. [52]. Source: own research

Three constructs exhibit a tendency towards high scores, such as perceived social norm, educational environment, and attitude towards entrepreneurship. Three constructs (perceived entrepreneurial capacity, perceived social norm, and entrepreneurial intention) present a tendency towards medium scores. The outer loadings between a construct and its indicators are bigger than 0.6. This indicates that the indicator reliability of all indicators support the convergent validity (Table 5).

**Table 5.** Indicator reliability.

| Reflective Model | Outer Loadings | *p*-Values |
|---|---|---|
| AE1 ← Attitude toward Entrepreneurship | 0.905 | 0.000 |
| AE2 ← Attitude toward Entrepreneurship | 0.924 | 0.000 |
| AE3 ← Attitude toward Entrepreneurship | 0.926 | 0.000 |
| AE4 ← Attitude toward Entrepreneurship | 0.883 | 0.000 |
| EE1 ← Educational Environment | 0.821 | 0.000 |
| EE2 ← Educational Environment | 0.898 | 0.000 |
| EE3 ← Educational Environment | 0.873 | 0.000 |
| EE4 ← Educational Environment | 0.923 | 0.000 |
| EI1 ← Entrepreneurial Intention | 0.813 | 0.000 |
| EI2 ← Entrepreneurial Intention | 0.719 | 0.000 |
| EI3 ← Entrepreneurial Intention | 0.874 | 0.000 |
| EI4 ← Entrepreneurial Intention | 0.883 | 0.000 |
| EI5 ← Entrepreneurial Intention | 0.916 | 0.000 |
| EI6 ← Entrepreneurial Intention | 0.830 | 0.000 |

**Table 5.** *Cont.*

| Reflective Model | Outer Loadings | *p*-Values |
|---|---|---|
| PEC1 ← Perceived Entrepreneurial Capacity | 0.885 | 0.000 |
| PEC2 ← Perceived Entrepreneurial Capacity | 0.886 | 0.000 |
| PEC3 ← Perceived Entrepreneurial Capacity | 0.765 | 0.000 |
| PEC4 ← Perceived Entrepreneurial Capacity | 0.896 | 0.000 |
| PSN1 ← Perceived Social Norm | 0.827 | 0.000 |
| PSN2 ← Perceived Social Norm | 0.880 | 0.000 |
| PSN3 ← Perceived Social Norm | 0.877 | 0.000 |

Source: SmartPLS' results.

Table 6 also shows all construct or latent fulfill convergent validity and internal consistency criteria. Convergent validity is suggested by an AVE greater than 0.5. The rho $_A$ and Cronbach's alpha score greater than 0.7 suggests composite reliability.

**Table 6.** Assessment of convergent validity and internal consistency.

| Latent Variables | rho $_A$ | CA | CR | AVE |
|---|---|---|---|---|
| Attitude toward Entrepreneurship | 0.931 | 0.930 | 0.950 | 0.828 |
| Educational Environment | 0.923 | 0.903 | 0.932 | 0.774 |
| Entrepreneurial Intention | 0.918 | 0.916 | 0.935 | 0.708 |
| Perceived Entrepreneurial Capacity | 0.884 | 0.880 | 0.918 | 0.739 |
| Perceived Social Norm | 0.829 | 0.826 | 0.896 | 0.743 |

Source: SmartPLS' results.

Table 7 shows that the HTMT ratio justifies the discriminant validity. The HTMT ratio of correlations are smaller than 0.85 or 0.90.

**Table 7.** Heterotrait–Monotrait (HTMT) ratio.

| Correlation | Ratio |
|---|---|
| Educational Environment → Attitude toward Entrepreneurship | 0.574 |
| Entrepreneurial Intention → Attitude toward Entrepreneurship | 0.703 |
| Entrepreneurial Intention → Educational Environment | 0.441 |
| Perceived Entrepreneurial Capacity → Attitude toward Entrepreneurship | 0.493 |
| Perceived Entrepreneurial Capacity → Educational Environment | 0.358 |
| Perceived Entrepreneurial Capacity → Entrepreneurial Intention | 0.779 |
| Perceived Social Norm → Attitude toward Entrepreneurship | 0.666 |
| Perceived Social Norm → Educational Environment | 0.564 |
| Perceived Social Norm → Entrepreneurial Intention | 0.666 |

Source: SmartPLS' results.

To test the hypotheses, bootstrapping was ran with 5000 bootstrap subsamples. The results are illustrated in Figure 2 and Tables 8 and 9.

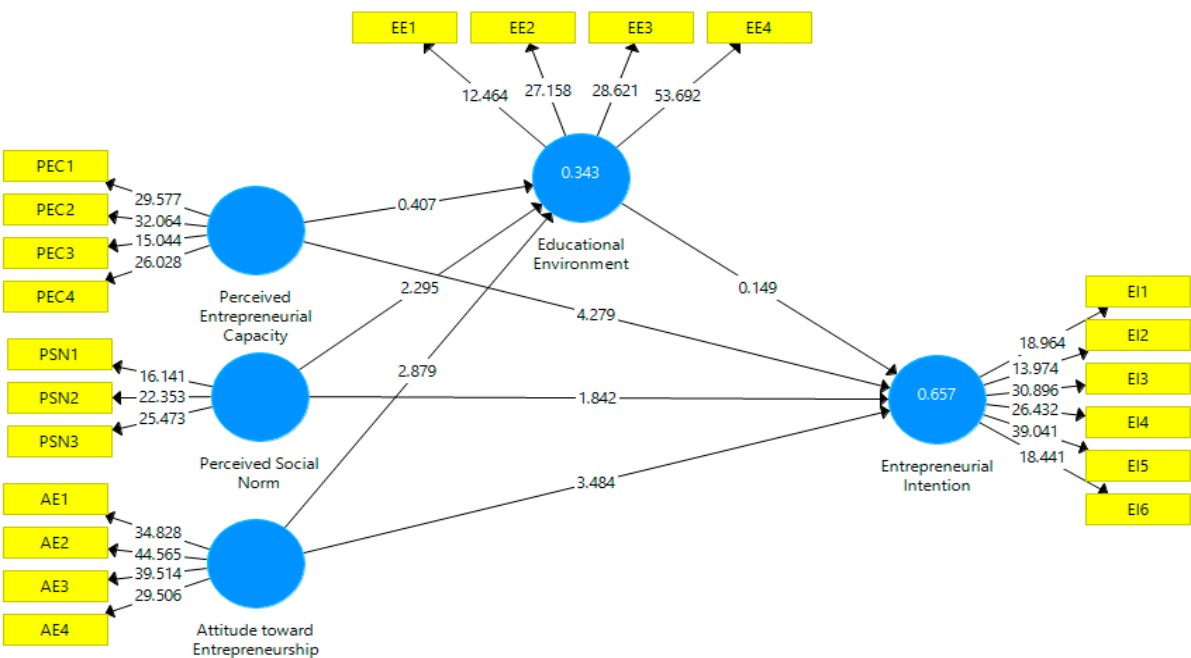

**Figure 2.** VB-SEM diagram output. Source: SmartPLS' results. Note: $R^2$ values for the educational environment and entrepreneurial intention are 0.343 and 0.657, respectively.

**Table 8.** Goodness-of-fit index.

| Latent Variables | AVE | $R^2$ |
| --- | --- | --- |
| Attitude toward Entrepreneurship | 0.828 | - |
| Educational Environment | 0.774 | 0.343 |
| Entrepreneurial Intention | 0.708 | 0.657 |
| Perceived Entrepreneurial Capacity | 0.739 | - |
| Perceived Social Norm | 0.743 | - |
| Average | 0.758 | 0.5 |
| GoF $= \sqrt{\text{AVE} \times R^2}$ | | 0.616 |

Source: own research.

**Table 9.** Direct and indirect effects.

| Direct/Indirect Path | β | T-Stat. | *p*-Values | Decision |
| --- | --- | --- | --- | --- |
| AE -> EE | 0.366 | 2.879 | 0.004 *** | Supported |
| AE -> EI | 0.346 | 3.484 | 0.000 *** | Supported |
| EE -> EI | −0.010 | 0.149 | 0.882 | Not Supported |
| PEC -> EE | 0.054 | 0.407 | 0.684 | Not Supported |
| PEC -> EI | 0.465 | 4.279 | 0.000 *** | Supported |
| PSN -> EE | 0.257 | 2.295 | 0.022 ** | Supported |
| PSN -> EI | 0.186 | 1.842 | 0.066 * | Supported |
| AE -> EE -> EI | −0.004 | 0.148 | 0.883 | Not Supported |
| PEC -> EE -> EI | −0.001 | 0.044 | 0.965 | Not Supported |
| PSN -> EE -> EI | −0.003 | 0.140 | 0.889 | Not Supported |

Note: * $p < 0.1$ level, ** $p < 0.05$ level, *** $p < 0.01$ level, respectively. Sources: SmartPLS' results.

Figure 2 demonstrates that the perceived social norm, perceived entrepreneurial capacity, and attitude toward entrepreneurship account for 34.3% (0.343) of the variation in the educational environment. Similarly, the perceived social norm, attitude toward entrepreneurship, perceived entrepreneurial capacity, and educational environment account for 65.7% (0.657) of the variation in entrepreneurial intention.

The goodness-of-fit index of 0.616 indicates a satisfactory model fit and considerable predictive power.

All of the indirect hypotheses are supported, except for the indirect hypothesis. Entrepreneurial intention is positively influenced by perceived entrepreneurial capacity ($\beta = 0.465$; $p < 0.01$). The perceived social norm influences entrepreneurial intention in a favorable way ($\beta = 0.465$; $p < 0.1$), and entrepreneurial intention is positively influenced by attitude toward entrepreneurship ($\beta = 0.346$; $p < 0.01$). The indirect effect does not support the proposed hypothesis ($\beta = -0.004$; $\beta = -0.001$; $\beta = -0.003$).

## 5. Discussion

This research study is to examine education's role in the development of entrepreneurial intention based on the theory of planned behavior. The research established four hypotheses and checked their validity. The findings demonstrate that entrepreneurial intention is positively correlated with each of the three factors (perceived entrepreneurial capacity, perceived social norm, and attitude toward entrepreneurship). Hypothesis 1 suggested that perceived entrepreneurial capacity has a positive influence on entrepreneurial intention [10,11,22]. Fear of failure is the factor which hampers the decision-making of whether to try [53]. As this research established, cognitive self-efficacy in entrepreneurship boosts confidence in starting a new business. Our findings are in line with Pihie and Bagheri and Boubker et al., who have stated that entrepreneurial intention is positively correlated with self-efficacy cognition [22,30]. Our findings are also supported by Lee et al. and Lingappa et al., who revealed that, when performing entrepreneurial tasks, self-efficacy is an important attribute [32,54].

Hypothesis 2 suggested that perceived social norm has a positive influence on entrepreneurial intention [10,11]. Research by Karimi et al. and Soetjipto et al. [10,11] supports our findings. Supportive surroundings in realizing entrepreneurship have a significant influence in developing entrepreneurial intention [55,56]. Entrepreneurial startups thrive in a particular social system with a culture that supports entrepreneurial activities [55,56]. As mentioned before, the Korean government has known of the unlimited potential of generating wealth in entrepreneurship and has arranged diverse policies to support entrepreneurial business [15]. The supportive social atmosphere facilitates building entrepreneurial intention. Hypothesis 3 suggested that attitude toward entrepreneurship has a positive influence on entrepreneurial intention. Our findings are supported by Karimi et al., Boubker et al., Boubker et al., and Soetjipto [2,10,11,22]. A positive attitude toward entrepreneurship serves as a catalyst for building entrepreneurial intention [36]. A cognitive of consideration of a certain thing offers strong influence in making a related decision and propels the decision into action [36]. The results definitely support Hypotheses 1, 2, and 3 and also support previous research papers [2,6,7,9,14,20,22–26,29,30].

However, the results do not support the mediating role of the educational environment in the relationships between each of the three factors and entrepreneurial intention. Hypothesis 4 suggested that the educational environment has an indirect effect in the relation of perceived entrepreneurial capacity, perceived social norm, and attitude toward entrepreneurship to entrepreneurial intention. This is different from the results of previous research papers [55,57–59]. This could be due to regional characteristics and to the different methods of analysis. Previous research papers were conducted in a certain country (Indonesia, India, United Kingdom, Finland, Spain, and Senegal). This study is also conducted in one specific country (Korea); moreover, further difference is generated in where the educational factor locates in the structure of the study model. Some previous research papers only checked the influence of education in a direct relationship with entrepreneurial

intention. This study set educational factor as a mediating factor [2,10,11,22]. This study tried to check the mediating role in the relationship. Seikkula-Leino et al. found that teachers' commitment in entrepreneurial education cause a difference in outcome, and that the commitment shows differently in each country (United Kingdom, Finland, and Spain). Each country's educational environment has been formed differently with its own particular situation [57]. Saadin and Daskin found that the difference in gender could give rise to a difference in entrepreneurial intention [60]. Females have more desire for entrepreneurship than males. Male accounts for about two-thirds of sample in this study. According to Saadin and Daskin, it could be assumed that the results of this research have some validity [60].

## 6. Conclusions

This research examines factors that are associated with entrepreneurial intention among students at Mirae High School of Science and Technology. We found that perceived of social norms, perceived entrepreneurial capacity, and attitudes toward entrepreneurship all have a favorable effect on entrepreneurial intention in the context of South Korea. Our theoretical foundation was developed from the theory of planned behavior [7], utilizing the educational environment as a mediating variable. By examining predictors of entrepreneurial intention, the first theoretical implication contributes to the literature on entrepreneurship [2,22]. Our research demonstrates how students' entrepreneurial intention is influenced by their perceived entrepreneurial capacity, social norms, and attitude; however, the educational setting is ineffective as a mediating factor in this relationship. A basic explanation for this is that the provided entrepreneurship model appears to focus on the theory rather than practice [11]. Another explanation is that the students came into the program with optimistic perspectives on entrepreneurship and had strong intentions to pursue it. There was less of a chance to influence their attitudes and intentions [10]. Second, the theory of planned behavior is concerned with how individual attitudes, subjective norms, and perceived behavioral control influence people's behavior [7]. This research also provides an essential recommendation for the practitioners. In practice, the outcomes would offer marvelous practical significance for refining high school students' entrepreneurial intentions, increasing the field of knowledge in relation to entrepreneurship among students, and providing students with entrepreneurial skills. Due to the fact that so few prior studies on entrepreneurial intention have concentrated on the mediating role of the educational environment, our research is unique. The intention to start a business is significantly influenced by perceptions of one's entrepreneurial potential, social norms, and attitude toward the field [10,11].

This research's findings add another empirical contribution related to the importance of the educational environment from the perspective of South Korean students to the literature review. It provides practical advice on how to teach students entrepreneurial skills. This research advances our understanding of the factors that predict students' entrepreneurial intention. Governments could use it to make decisions about entrepreneurship.

The research's methodological implication is the utilization of the bootstrapping approach [61] in VB-SEM or PLS-SEM. In order to compensate for the issue of the small sample size, bootstrapping techniques were used. Along with its advantages, this research has a number of limitations. First, a larger sample size is needed to increase the model's validity even though the sample size is adequate enough to test the structural model [42,43]. Second, the additional regions of South Korea should conduct tests in order to increase the predictors' applicability. Third, it may be important to determine how long the school has been teaching students entrepreneurial skills in order to gauge actual student entrepreneurial intention. To increase research validity, it is strongly advised to conduct a longitudinal study that takes into account the effects of time series and the cross-sectional scope of entrepreneurial intention. Lastly, an expanded understanding of students' entrepreneurial behavior may result from the analysis of various predictors of the entrepreneurial intention model.

**Author Contributions:** Conceptualization, M.-S.K.; methodology, A.D.H.; software, C.-W.L.; validation, M.-S.K.; formal analysis, A.D.H.; investigation, C.-W.L.; resources, M.-S.K.; data curation, C.-W.L.; writing—original draft preparation, A.D.H.; writing—review and editing, M.-S.K.; visualization, C.-W.L.; supervision, M.-S.K.; project administration, A.D.H.; funding acquisition, M.-S.K. All authors have read and agreed to the published version of the manuscript.

**Funding:** This research received internal funding from Chung Yuan Christian University-Taiwan.

**Institutional Review Board Statement:** Not applicable.

**Informed Consent Statement:** Not applicable.

**Data Availability Statement:** Not applicable.

**Conflicts of Interest:** The authors declare no conflict of interest.

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
