# Peer review of "Predictors of Entrepreneurial Intention among High School Students in South Korea"

_sustainability, doi:10.3390/su142114168_

Round 1

Reviewer 1 Report

Abstract

1.    The paper’ contribution needs to be stated more clearly.

Introduction

1.    Specify what makes this article different from the rest of studies that are available in the literature.

2.    Identify the gap in exiting literature, by arguing what is missing or inadequate in existing solutions and thus your study is necessary. This needs to be briefly noted in Introduction, and then further elaborated in the Literature Review, with in-depth analysis and substantiation of citations.

Literature Review

1.    Add some literature related to the main research questions.

2.    The context of the research is not explicitly established. The paper takes it for granted that the readers share the authors’ view.

3.    Before developing the conceptual framework of the current research, the paper should first engage with existing theoretical frameworks in literature to demonstrate the need for the proposed framework. What literature is there to support this claim? This aspect of the paper is one of the weakest and needs in-depth relook to strengthen the theoretical aspect.

4.    It would be helpful in clarifying the importance of the proposed study if the paper can include references to some latest articles published in recent years within the scope of the current research. The literature review should be more carefully synthesized and structured.

5.     

Discussion of Results/Findings

1.    Discussion of results/findings, needs to related to previous literature and compare and contrast the findings/claims against that of previous studies

Conclusion

1.    The contribution of your study needs to be clearly articulated in this section. The contribution is not clear and need to be stated.

2.    The section needs to include some recommendations for practitioners based on the findings, if appropriate.

----------------------------

Finally, the paper needs minor proofreading and editing to clear typos and grammatical anomalies.

/* Similarity check with iThenticate revealed a similarity index of 21% which is considered NOT appropriate. A maximum of around 60 quoted words is accepted per paper. There are 4 papers with over 60 words. No previously copyrighted material can be used. See the attachment file with the plagiarism check results.

/* In preparing a revised manuscript, please also include a table of how you have responded to each of the issues listed above point by point.

Dear AUTHOR, summarizing my feedback, I expect your contribution to be highly valued by journals readers, if you improve it according to the review statements. I enjoyed reading your manuscript, and I am looking forward to reading your revisions and seeing your research published. 

Best wishes

Author Response

We are pleased to resubmit for possible publication, the revised version of manuscript ID: sustainability-2001671 entitled “Predictors of Entrepreneurial Intention Among High School Students in South Korea” We would also like to thank the reviewer 1 for the insightful and helpful comments on the manuscript. We have addressed reviewer 1 comments as outlined below (Yellow Highlighted):

Reviewer 1

1.

COMMENT:

Abstract

The paper’ contribution needs to be stated more clearly.

RESPONSE: 

We are thankful for reviewer’s insightful and helpful comments.

ACTION:

Abstract: The purpose of this research is to gauge relationships between perceived entrepreneurial capacity, perceived social norm, attitude toward entrepreneurship, and entrepreneurial intention drawn on the theory of planned behavior and to test the mediating role of the educational environment in the relationships. Based on the conceptual structure of the theory, this research was organized to explore how the conception of the theory works in the event of Korean students and to further check the role of the educational environment in the course of its working. At Mirae High School of Science and Technology, eighty-nine respondents were used for the statistical analysis. One model was set to manifest this research concept. This research employs the Variance Based - Structural Equation Modeling (VB-SEM). All of the direct hypotheses are supported except the indirect hypothesis. The perceived entrepreneurial capacity positively influences entrepreneurial intention. The perceived social norm positively influences entrepreneurial intention. And the attitude towards entrepreneurship positively influences entrepreneurial intention. The indirect effect is not in line with the expected hypothesis. The research’s findings contribute to the literature review by adding another empirical confirmation (educational environment's role) from South Korean students’ viewpoint. It offers a useful information of providing students with entrepreneurial skills. This research also contributes to the advancement of knowledge in relation to the predictors of entrepreneurship in students. It would help governments to make decisions on entrepreneurship. It would show the direction to where government or policymaker leads entrepreneurship and its education.

2.

COMMENT:

Introduction

1.     Specify what makes this article different from the rest of studies that are available in the literature.

2.     Identify the gap in exiting literature, by arguing what is missing or inadequate in existing solutions and thus your study is necessary. This needs to be briefly noted in Introduction, and then further elaborated in the Literature Review, with in-depth analysis and substantiation of citations.

RESPONSE: 

We appreciate the reviewer’s insightful and helpful comments.

ACTION:

As entrepreneurship takes on economic importance, the education specialized for nurturing entrepreneurship has been focused. Recent years have seen an increasing discussion about entrepreneurship education from various aspects. Many related research papers have been dominated by a focus on undergraduate or graduate business education [17–23]. And this paper puts educational environment as mediator in existing structure of theory of planned behavior. Given these considerations, this research is certainly welcome.

Recent literatures:

Aliedan, M.M.; Elshaer, I.A.; Alyahya, M.A. Influences of University Education Support on Entrepreneurship Orientation and Entrepreneurship Intention: Application of Theory of Planned Behavior. Sustainability 2022, 14, 13097, doi:10.3390/su141911890.

Botezat, E.A.; Constăngioară, A.; Dodescu, A.O.; Pop-Cohuţ, I.C. How Stable Are Students’ Entrepreneurial Intentions in the COVID-19 Pandemic Context? Sustainability 2022, 14, 5690, doi:10.3390/su14095690.

Ge, X.; Wang, J. A Meta-Analysis of the Relationship between Financing Efficiency and Entrepreneurial Vitality: Evidence from Chinese College Students. Sustainability 2022, 14, 10826, doi:10.3390/su141710826.

Hatos, R.; Cioban, S.; Bea, G.; Dodescu, A.; Hatos, A. Assessing the Impact of Entrepreneurial Education on Entrepreneurial Intentions among Romanian Doctoral Students and Postdoctoral Researchers. Sustainability 2022, 14, 8369, doi:10.3390/su14148369.

Li, Z.; Zhang, W.; Zhou, Y.; Kang, D.; Feng, B.; Zeng, Q.; Xu, L.; Zhang, M. College Students’ Entrepreneurial Intention and Alertness in the Context of the COVID-19 Pandemic. Sustainability 2022, 14, 7713, doi:10.3390/su14137713.

Zhang, W.; Li, Y.; Zeng, Q.; Zhang, M.; Lu, X. Relationship between Entrepreneurship Education and Entrepreneurial Intention among College Students: A Meta-Analysis. International Journal of Environmental Research and Public Health 2022, 19, 12158, doi:10.3390/ijerph191912158.

3.

COMMENT:

Literature Review

1.     Add some literature related to the main research questions.

2.     The context of the research is not explicitly established. The paper takes it for granted that the readers share the authors’ view.

3.     Before developing the conceptual framework of the current research, the paper should first engage with existing theoretical frameworks in literature to demonstrate the need for the proposed framework. What literature is there to support this claim? This aspect of the paper is one of the weakest and needs in-depth relook to strengthen the theoretical aspect.

4.     It would be helpful in clarifying the importance of the proposed study if the paper can include references to some latest articles published in recent years within the scope of the current research. The literature review should be more carefully synthesized and structured.

RESPONSE: 

We thanked for reviewer’s insightful and helpful suggestions.

ACTION:

Self-efficacy is a crucial characteristic in exercising entrepreneurial tasks, and entrepreneurial intention is linked to the ability [1,2].

Dragin et al. found that family characteristics and tradition affect greatly in shaping one’s entrepreneurial intention [3].

A favorable social support strengthening the willingness of being an entrepreneur [4].

Sobaih and Elshaer found that gender role built out of social custom also determines forming characteristics favorable to entrepreneurship intention [5].

Aliedan et al. found that undergraduate education supports building entrepreneurship. They noted that entrepreneurial education is vital to development of entrepreneurial intention in both a direct way and an indirect way. It shows that entrepreneurial intention hinges on level of education and its curriculum [6]. Chang et al. found that entrepreneurial behavior originated from intention, mindset, and competency, and that three are designed by educational setting [7]. Many recent research papers prove educational role in forming entrepreneurship [8,9]. Recent scholarly effort shows that entrepreneurship cannot be realized without educational setting [10–12].

Recent Literatures:

Lee, S.; Kang, M.J.; Kim, B.K. Factors Influencing Entrepreneurial Intention: Focusing on Individuals’ Knowledge Exploration and Exploitation Activities. Journal of Open Innovation: Technology, Market, and Complexity 2022, 8, 165, doi:10.3390/joitmc8030165.

Lingappa, A.K.; Kamath, A.; Mathew, A.O. Engineers and Social Responsibility: Influence of Social Work Experience, Hope and Empathic Concern on Social Entrepreneurship Intentions among Graduate Students. Social Sciences 2022, 11, 430, doi:10.3390/socsci11100430.

Dragin, A.S.; Mijatov, M.B.; Ivanović, O.M.; Vuković, A.J.; Džigurski, A.I.; Košić, K.; Knežević, M.N.; Tomić, S.; Stankov, U.; Vujičić, M.D.; et al. Entrepreneurial Intention of Students (Managers in Training): Personal and Family Characteristics. Sustainability 2022, 14, 7345, doi:10.3390/su14127345.

Si, W.; Yan, Q.; Wang, W.; Meng, L.; Zhang, M. Research on the Influence of Non-Cognitive Ability and Social Support Perception on College Students’ Entrepreneurial Intention. International Journal of Environmental Research and Public Health 2022, 19, 11981, doi:10.3390/ijerph191911981.

Sobaih, A.E.E.; Elshaer, I.A. Structural Equation Modeling-Based Multi-Group Analysis: Examining the Role of Gender in the Link between Entrepreneurship Orientation and Entrepreneurial Intention. mathematics 2022, 10, 3719, doi:10.3390/math10203719.

Aliedan, M.M.; Elshaer, I.A.; Alyahya, M.A. Influences of University Education Support on Entrepreneurship Orientation and Entrepreneurship Intention: Application of Theory of Planned Behavior. Sustainability 2022, 14, 13097, doi:10.3390/su141911890.

Chang, A.; Chang, D.; Chen, T.-L. Detecting Female Students Transforming Entrepreneurial Competency, Mindset, and Intention into Sustainable Entrepreneurship. Education Studies 2022, 14, 12970, doi:10.3390/su142012970.

Lin, S.; De-Pablos-Heredero, C.; Botella, J.L.M.; Lin-Lian, C. Entrepreneurial Intention of Chinese Students Studying at Universities in the Community of Madrid. Sustainability 2022, 14, 5475, doi:10.3390/su14095475.

Nitu-Antonie, R.D.; Stamenovic, K.; Brudan, A. A Moderated Serial – Parallel Mediation Model of Sustainable Entrepreneurial Intention of Youth with Higher Education Studies in Romania. Sustainability 2022, 14, 13342, doi:10.3390/su142013342.

Ragazou, K.; Passas, I.; Garefalakis, A.; Kourgiantakis, M.; Xanthos, G. Youth’s Entrepreneurial Intention: A Multinomial Logistic Regression Analysis of the Factors Influencing Greek HEI Students in Time of Crisis. Sustainability 2022, 14, 13164, doi:10.3390/su142013164.

Zhang, L.; Fu, Y.; Wei, Y.; Chen, H.; Xia, C.; Cai, Z. Predicting Entrepreneurial Intention of Students: Kernel Extreme Learning Machine with Boosted Crow Search Algorithm. Applied Sciences 2022, 12, 6907, doi:10.3390/app12146907.

Zhu, R.; Zhao, G.; Long, Z.; Huang, Y.; Huang, Z. Entrepreneurship or Employment? A Survey of College Students’ Sustainable Entrepreneurial Intentions. Sustainability 2022, 14, 5466, doi:10.3390/su14095466.

4.

COMMENT:

Discussion of Results/Findings

1.     Discussion of results/findings, needs to related to previous literature and compare and contrast the findings/claims against that of previous studies

RESPONSE: 

We thanked for reviewer’s insightful and helpful suggestions.

ACTION:

Hypothesis 1 suggested that perceived entrepreneurial capacity has a positive influence on entrepreneurial intention [10,11,26]. Fear of failure is the factor which hampers the decision-making of whether to try [61]. As this research checked, cognitive self-efficacy in entrepreneurship boosts confidence in starting new business. Our findings are in line with Pihie and Bagheri and Boubker et al. which stated that entrepreneurial intention is positively correlated with self-efficacy cognition [26,34]. Our findings also supported by Lee et al. and Lingappa et al. which revealed that when performing entrepreneurial tasks, self-efficacy is an important attribute [36,37].

Hypothesis 2 suggested that perceived social norm has a positive influence on en-trepreneurial intention [10,11]. Research by Karimi et al. and Soetjipto et al. [10,11] supports our findings. Supportive surroundings in realizing entrepreneurship have a significant influence in building entrepreneurial intention [62,63]. A particular culture governing social system, whose culture is encouraging entrepreneurial activities, allows of a new try [62,63]. As mentioned before, Korean government has known unlimited potential of generating wealth in entrepreneurship and arranged diverse policies to support entrepreneurial business [15]. The supportive social atmosphere facilitates building the intention. Hypothesis 3 suggested that attitude toward entrepreneurship has a positive influence on entrepreneurial intention. Our findings supported by Karimi et al., Boubker et al., Boubker et al., and Soetjipto [2,10,11,26]. A positive attitude toward entrepreneurship serves as a catalyst for building entrepreneurial intention [41,64]. A cognitive of consideration about a certain thing gives its strong influence in making the related decision and propels the decision into action [41]. The results definitely support hypotheses 1, 2, and 3. And it also supports previous research papers [2,6,30,33,34,7,9,14,24,26–29].

But the results do not support the mediating role of educational environment in relationships between each of three factors and entrepreneurial intention. Hypothesis 4 suggested that educational environment has its indirect effect in the relation of perceived entrepreneurial capacity, perceived social norm, and attitude toward entrepreneurship to entrepreneurial intention. It has a difference from results of previous research papers [62,65–67].

5.

COMMENT:

Conclusion

1.     The contribution of your study needs to be clearly articulated in this section. The contribution is not clear and need to be stated.

2.     The section needs to include some recommendations for practitioners based on the findings, if appropriate.

RESPONSE: 

We thanked for reviewer’s insightful and helpful suggestions.

ACTION:

The research would also provide an essential recommendation for the practitioners. In practice, the outcomes would offer marvelous practical significance for refining high school students’ entrepreneurial intentions, increasing the field of knowledge in relation to entrepreneurship among students, and providing students with entrepreneurial skills.

The research's findings add another empirical contribution related to the importance of the educational environment from the perspective of South Korean students to the literature review. It provides practical advice on how to teach students entrepreneurial skills. This research advances our understanding of the factors that predict students' entrepreneurial intentions. Governments could use it to make decisions about entrepreneurship.

6.

ADDITIONAL COMMENTS:

1.     Finally, the paper needs minor proofreading and editing to clear typos and grammatical anomalies.

2.     Similarity check with iThenticate revealed a similarity index of 21% which is considered NOT appropriate. A maximum of around 60 quoted words is accepted per paper. There are 4 papers with over 60 words. No previously copyrighted material can be used. See the attachment file with the plagiarism check results.

3.     In preparing a revised manuscript, please also include a table of how you have responded to each of the issues listed above point by point.

4.     Dear AUTHOR, summarizing my feedback, I expect your contribution to be highly valued by journal’s readers, if you improve it according to the review statements. I enjoyed reading your manuscript, and I am looking forward to reading your revisions and seeing your research published.

RESPONSE: 

We thanked for reviewer’s insightful and helpful suggestions.

ACTION:

We have proofread the manuscript and paraphrase the sentences to reduce the similarity.

Reviewer 2 Report

The article is relevant to the mission of the journal. I consider the work to be relevant for several reasons. It is a study that contributes to increasing the field of knowledge in relation to entrepreneurship among students. 2. It is of vital importance in providing students with entrepreneurial skills.

The topic of the article "Predictors of Entrepreneurial Intention Among High School Students in South Korea" is interesting and a timely study as it is an emerging research problem in relation to the predictors of entrepreneurial intention.

The paper is well structured, facilitating the understanding of the study. The theoretical foundation is based on the research questions, providing current and new literature in relation to the study problem and the objectives set out.

Objective: The research problem and the objective of the study are well defined.

Method: A cross-sectional research using questionnaires.

The research steps are presented in a clear and structured way and the research questions are answered in a clear and detailed manner.

Results: This evaluator considers the results shown to be relevant to the study problem.

The study clearly specifies the limitations and lines of future research.

In short, I consider that this is a good work that will contribute to the advancement of knowledge in relation to the predictors of entrepreneurship in students, and will also help governments when making decisions on entrepreneurship.

I think the authors should include a section on the limitations of the study.

Author Response

We are pleased to resubmit for possible publication, the revised version of manuscript ID: sustainability-2001671 entitled “Predictors of Entrepreneurial Intention Among High School Students in South Korea” We would also like to thank the reviewer 2 for the insightful and helpful comments on the manuscript. We have addressed reviewer 2 comments as outlined below (Yellow Highlighted):

Reviewer 2

1.

COMMENT:

The article is relevant to the mission of the journal. I consider the work to be relevant for several reasons. It is a study that contributes to increasing the field of knowledge in relation to entrepreneurship among students. 2. It is of vital importance in providing students with entrepreneurial skills.

RESPONSE: 

We are thankful for reviewer’s insightful and helpful comments.

2.

COMMENT:

The topic of the article "Predictors of Entrepreneurial Intention Among High School Students in South Korea" is interesting and a timely study as it is an emerging research problem in relation to the predictors of entrepreneurial intention.

RESPONSE: 

We appreciate the reviewer’s insightful and helpful comments.

3.

COMMENT:

The paper is well structured, facilitating the understanding of the study. The theoretical foundation is based on the research questions, providing current and new literature in relation to the study problem and the objectives set out.

RESPONSE: 

We thanked for reviewer’s insightful and helpful suggestions.

4.

COMMENT:

Objective: The research problem and the objective of the study are well defined.

RESPONSE: 

We thanked for reviewer’s insightful and helpful suggestions.

5.

COMMENT:

Method: A cross-sectional research using questionnaires.

RESPONSE: 

We thanked for reviewer’s insightful and helpful suggestions.

6.

COMMENT:

The research steps are presented in a clear and structured way and the research questions are answered in a clear and detailed manner.

Results: This evaluator considers the results shown to be relevant to the study problem.

The study clearly specifies the limitations and lines of future research.

In short, I consider that this is a good work that will contribute to the advancement of knowledge in relation to the predictors of entrepreneurship in students, and will also help governments when making decisions on entrepreneurship.

I think the authors should include a section on the limitations of the study.

RESPONSE: 

We thanked for reviewer’s insightful and helpful suggestions.

ACTION:

Along with its advantages, this research has a number of limitations. First, although the sample size is sufficient to test the structural model [50,51], a larger sample size is required to improve the model's validity. Second, the predictors of students' entrepreneurial intentions should be tested in other areas of South Korea to improve their applicability. Third, it may be important to determine how long the school has been teaching students entrepreneurial skills in order to gauge actual student entrepreneurial intentions.

Reviewer 3 Report

Dear authors,

The article entitled Predictors of Entrepreneurial Intention Among High School Students in South Korea is interesting because it tackles a subject of high present interest.  

The article contains the appropriate structure. It is correctly divided into relevant sections and their content coincides with their titles. Bibliography is correctly formulated.

 The language of the article is mature, correct, adequate. English style is good.

1.      The abstract clearly specifies the objectives of the study

2.      The introduction is well implemented.

3.      The theoretical background in Literature Review contains enough literature on the addressed topic

4.      The Research Methodology chapter explains well the methodological framework

5.       The results are clearly formulated, the explanations are pertinent

6.      The conclusions are coherent are relevant, underlying the novelty of the paper.

Author Response

We are pleased to resubmit for possible publication, the revised version of manuscript ID: sustainability-2001671 entitled “Predictors of Entrepreneurial Intention Among High School Students in South Korea” We would also like to thank the reviewer 3 for the insightful and helpful comments on the manuscript. We have addressed reviewer 3 comments as outlined below:

Reviewer 3

1.

GENERAL COMMENTS:

The article is relevant to the mission of the journal. I consider the work to be relevant for several reasons. It is a study that contributes to increasing the field of knowledge in relation to entrepreneurship among students. 2. It is of vital importance in providing students with entrepreneurial skills.

The article entitled “Predictors of Entrepreneurial Intention Among High School Students in South Korea” is interesting because it tackles a subject of high present interest. 

The article contains the appropriate structure. It is correctly divided into relevant sections and their content coincides with their titles. Bibliography is correctly formulated.

 The language of the article is mature, correct, adequate. English style is good.

RESPONSE: 

We are thankful for reviewer’s insightful and helpful comments.

2.

COMMENT:

The abstract clearly specifies the objectives of the study

RESPONSE: 

We appreciate the reviewer’s insightful and helpful comments.

3.

COMMENT:

The introduction is well implemented

RESPONSE: 

We thanked for reviewer’s insightful and helpful suggestions.

4.

COMMENT:

The theoretical background in Literature Review contains enough literature on the addressed topic

RESPONSE: 

We thanked for reviewer’s insightful and helpful suggestions.

5.

COMMENT:

The Research Methodology chapter explains well the methodological framework

RESPONSE: 

We thanked for reviewer’s insightful and helpful suggestions.

6.

COMMENT:

The results are clearly formulated, the explanations are pertinent.

RESPONSE: 

We thanked for reviewer’s insightful and helpful suggestions.

7.

COMMENT:

The conclusions are coherent are relevant, underlying the novelty of the paper.

RESPONSE: 

We thanked for reviewer’s insightful and helpful suggestions.

Round 2

Reviewer 1 Report

The revised version improves the first. Also, the plagiarism ratio is 16% now. So endorse the publication.